# Deep Sequencing of Small RNA Reveals the Molecular Regulatory Network of *AtENO2* Regulating Seed Germination

**DOI:** 10.3390/ijms22105088

**Published:** 2021-05-11

**Authors:** Yu Wu, Lamei Zheng, Jie Bing, Huimin Liu, Genfa Zhang

**Affiliations:** Beijing Key Laboratory of Gene Resource and Molecular Development, College of Life Sciences, Beijing Normal University, Beijing 100875, China; wuyu@mail.bnu.edu.cn (Y.W.); zhenglm100@163.com (L.Z.); bingjie@bnu.edu.cn (J.B.); Huimin_Liu521@163.com (H.L.)

**Keywords:** seed germination, *AtENO2*, miRNAs, deep sequencing, regulation network

## Abstract

Seed germination is a key step in the new life cycle of plants. In agriculture, we regard the rapid and consistent process of seed germination as one of the necessary conditions to measure the high quality and yield of crops. ENO2 is a key enzyme in glycolysis, which also plays an important role in plant growth and abiotic stress responses. In our study, we found that the time of seed germination in *A**tENO2* mutation (*eno2^−^*) was earlier than that of wild type (WT) in *Arabidopsis thaliana*. Previous studies have shown that microRNAs (miRNAs) were vital in seed germination. After deep sequencing of small RNA, we found 590 differentially expressed miRNAs in total, of which 87 were significantly differentially expressed miRNAs. By predicting the target genes of miRNAs and analyzing the GO annotation, we have counted 18 genes related to seed germination, including *ARF* family, *TIR1*, *INVC*, *RR19*, *TUDOR2*, *GA3OX2*, *PXMT1*, and *TGA1*. MiR9736-z, miR5059-z, ath-miR167a-5p, ath-miR167b, ath-miR5665, ath-miR866-3p, miR10186-z, miR8165-z, ath-miR857, ath-miR399b, ath-miR399c-3p, miR399-y, miR163-z, ath-miR393a-5p, and ath-miR393b-5p are the key miRNAs regulating seed germination-related genes. Through KEGG enrichment analysis, we found that phytohormone signal transduction pathways were significantly enriched, and these miRNAs mentioned above also participate in the regulation of the genes in plant hormone signal transduction pathways, thus affecting the synthesis of plant hormones and further affecting the process of seed germination. This study laid the foundation for further exploration of the *AtENO2* regulation for seed germination.

## 1. Introduction

Seed germination is a complex multi-step process of transforming dormant seeds into seedlings with high metabolic activity. It is a key step in the plant life cycle [1]. In agricultural production, the rapid and uniform germination process of seeds is usually regarded as one of the necessary conditions to measure the quality and high yield of crops. The factors affecting germination mainly include internal genetic factors and external environmental factors [2,3,4]. More and more studies have shown that phytohormones take a significant role in the process of seed germination. Abscisic acid (ABA) is important in seed development by inducing seed dormancy to inhibit seed germination [5]. However, gibberellin (GA) and ABA play an antagonistic role in seed dormancy and germination, mainly inhibiting seed dormancy and promoting germination [6,7,8]. Exogenous low concentration of auxin (IAA) can promote germination of *Arabidopsis* seeds [9]; on the contrary, high concentration can inhibit germination [10]. Studies have shown that *etr (Ethylene resistant)* mutants promote the accumulation of ABA and lower the seed germination rate, which also reflects the negative regulatory effect of ethylene on germination [11,12]. Brassinolide (BR), which has attracted much attention in recent years, also plays a unique role in germination. Compared with the WT, BR insensitive mutant *bri* has a lower germination rate, but it can finally germinate without exogenous BR [13,14,15]. However, the specific mechanism needs to be further studied.

Meanwhile, with the continuous progress of science, technology, and research methods in recent years, more and more genes have been shown to play a crucial role in the process of seed germination, such as enolase (EC 4.2.1.11), which is also called 2-phospho-D-glycerate hydrolase, and is shortened as ENO. ENO is a rate-limiting enzyme in glycolysis. It catalyzes the conversion of 2-phosphate-d-glyceric acid (2-PGA) to phosphoenolpyruvic acid (PEP), and in the process of gluconeogenesis, it also catalyzes the reverse reaction of the reaction [16]. In plants, ENO1, ENO2, and ENO3 are the main isozymes, and ENO2 has higher enzyme activity than ENO1 and ENO3 [17]. In recent years, we found that the *ENO2* gene not only plays an important role in glycolysis but also is indispensable in plant response to abiotic stress and normal growth and development [18,19,20,21]. Our lab obtained homozygous mutants of the *AtENO2* gene (*eno2^−^*) through continuous screening and identification. Compared with WT, *eno2^−^* showed shorter main root, fewer lateral roots, smaller cotyledons, shorter plant, smaller rosette leaves, poor pollen development, and significantly shorter pods. Recently, we found that after the *AtENO2* gene mutation, the germination rate of seed was faster than that of WT. With the continuous development of high-throughput sequencing, the important role of miRNAs in germination has gradually become well known [22,23]. Based on this, we want to further explore the molecular mechanism of *AtENO2* gene-regulating seed germination from the miRNA level.

MiRNAs are about 19–24 nucleotides (nt) in length. They regulate gene expression by targeting mRNAs for cleavage or translation [24]. The miRNAs play diverse roles in various aspects of plant development, including leaf polarity [25,26], reproduction and root development [27], and flower development [28,29]. Previous studies have shown that miRNAs can be involved in regulating seed germination [22]. After mutations of key genes for small RNA synthesis, such as *DCL1*, *HYL1*, *HEN1*, and *AGO1*, serious defects in embryogenesis and seed development occurred [30]. In *dcl1* mutant, seed germination time of plants was earlier than that of WT [30]. Many miRNAs, such as miR417, miR402, miR165/166, miR160, miR167, miR156, and miR159, have been found to regulate seed germination. Besides, miR417 negatively regulates seed germination in *Arabidopsis thaliana* under salt stress [31]. Similarly, overexpression of miR402 promoted seed germination and seedling growth of *Arabidopsis* under salt, dehydration, or low-temperature stress [32]. Using deep sequencing, we found most of the miR156 family members were mainly expressed in embryos. The increase of miR156 level in mature embryos can inhibit the expression of target genes *SPL3,4,5,* keep seeds dormant, and inhibit seed germination [33]. In *Arabidopsis*, *ARF10* is the target gene of miR160. The negative regulation of miR160 on *AEF10* plays an important role in seed germination and after germination [9]. The miR159 regulates the expression of *GAMYB* mRNA, and GAMYB protein acts as a negative regulator of GA signaling cascade during seed germination [7,34]. Additionally, miR159 regulates transcription factors *MYB33* and *MYB101*, which are positive regulators of ABA signal transduction during seed dormancy and germination [35]. This indicates that miR159 is involved in the dynamic seed germination process by regulating GA and ABA hormone signal transduction. All these results revealed that miRNA is vital in seed germination. However, up to now, there is no report about *AtENO2* regulating seed germination at the miRNA level.

In this study, we aimed to identify miRNAs differentially expressed in WT and *eno2^−^*, and to identify their molecular mechanisms in regulating seed germination. For this reason, we performed miRNA deep sequencing of WT, and *eno2^−^* seedlings were grown for one week. We identified 590 differentially expressed miRNAs (DEMs); among them, 87 miRNAs showed more than a 2-fold change in expression and the *p*-value was less than 0.05 (log2|FC| ≥ 1, *p* ≤ 0.05). We also identified seven target genes related to seven miRNA families, which is essential for regulating seed germination.

## 2. Results

### 2.1. Statistical Analysis on Germination of eno2^−^

*ENO2* plays a key role in glycolysis, and it also participates in the process of affecting the growth and development of plants and responding to abiotic stress. In this study, the *eno2^−^* (T-DNA insertion site can be seen in Appendix A) screened and identified in our laboratory were used as materials to calculate the germination status of WT and *eno2^−^*. In production, we regard the rapid and uniform germination process of seeds as one of the necessary conditions to measure the high quality and high yield of crops. Generally, the ratio of the number of seeds germinated on the third day of growth to the total number of seeds sown is regarded as germinability, and the statistical results on the seventh day are regarded as the germination rate. After statistical analysis, we found that the mutation of the *A**tENO2* gene was faster than that of WT in the first three days of germination, but there was no significant difference in the later stage of germination (Figure 1). That is, the germinability of the mutant is higher than that of the WT, but there is no significant difference in germination rate between them. Usually, when the germination rate is the same, high germinability indicates strong seed vitality, neat germination, and consistent seedling emergence. This result also shows that the *AtENO2* does play an indispensable role in the germination of *Arabidopsis thaliana*, and the specific regulatory factors need to be further explored.

### 2.2. Data Mining of Small RNA Sequencing

To explore the effect of *eno2^−^* on the miRNA level, we constructed two small RNA libraries of WT (Col-0) and *eno2^−^* for the following research. After high-throughput sequencing, 9981252 and 10369769 clean reads were obtained from WT and *eno2^−^* libraries, respectively. After filtering and screening, low-quality reads without 3′ junction, reads with 5′ junction, and poly A were removed. Finally, we obtained 9661299 and 10071178 small RNA clean tags sequences, which can be used for our research (Appendix A). The length of small RNA is mainly distributed between 16–28 nt, and the distribution of small RNA with different lengths is also different. The number of small RNAs distributed between 20–24 nt is the largest, of which 21 nt is the most abundant, followed by 24 nt and 22 nt, accounting for 24.7% and 26.7%, respectively (Figure 2). Between 16–20 nt, the number of small RNAs in WT are more than *eno2^−^*. On the contrary, in the range of 21–25 nt, *eno2^−^* are more than WT. Especially, at the 24 nt peak of the mutant, it is significantly higher than WT.

We abbreviated small RNA sequence as tag, and we also compared all tags with various RNAs, summarized the annotation situation, then classified all tag sequences in the order of priority rRNA, etc. (rRNA, scRNA, snRNA, snoRNA, tRNA) > miRNA, etc. (exist_miRNA, conserved_miRNA, novel_miRNA) > repeat > exon > intron. If the tag does not match any annotation information, it will be expressed as unann. For the two mRNA libraries, the number of tag species was 1053660 and 1017006, and the tag abundance was 9661299 and 10071178 in WT and *eno2^−^*, respectively (Table 1). It can also be shown that small RNA is crucial in regulating plant growth and development, and the *ENO2* gene also has a certain effect on the tag abundance of small RNA. Additionally, we counted the number of miRNA in WT and *eno2^−^*, and the results were shown by veen diagram. The results showed that the number of existing miRNAs in WT and *eno2^−^* was 243 and 247, and 232 are in common; while that of conserved miRNA was 175 and 173, respectively, and 132 are in common. Besides, combined with the reference sequence, we used the hairpin structure to predict novel miRNA, and the number of new miRNAs identified in WT and *eno−^-^* was 112 and 115, respectively, and 111 are in common (Figure 3). In the future, we further explored and analyzed the data to clarify the effect of *eno2^−^* on small RNA level, and the mechanism of affecting seed germination.

### 2.3. Expression Analysis of miRNA and Prediction of Key Target Genes

To understand more details of the effect of *AtENO2* on miRNA, we counted 590 (DEMs) in total between WT and *eno2^−^* (Appendix A). The volcano graph can clearly and intuitively display the expression of miRNAs between samples. Based on this, we drew a volcano map of 590 DEMs that were screened (Figure 4). According to the change in expression, we screened the differential expression miRNAs whose expression changed by more than 2 times, and the *p*-value was less than 0.05; a total of 87 (Figure 5 and Appendix A), that is, significantly DEMs, of which 52 miRNAs were significantly up-regulated expression, and 35 miRNAs were significantly down-regulated. MiRNAs, such as novel-m0063-5p, novel-m0088-3p, miR5020-z, ath-miR5665, miR163-z, and novel-m0108-5p have been significantly up-regulated. Their functions need to be further explored. The miRNAs, such as ath-miR771, ath-miR8176, novel-M0111-5P, miR10186-Z, ath-miR399b, ath-miR399C-3P, miR5059-z, miR8165-z, miR399-Y, miR10186-Z, and miR395-X were significantly down-regulated.

By predicting the target genes of DEMs (Appendix A), we found 18 genes related to seed germination, including the *ARF* family, *TIR1*, *INVC*, *RR19*, *GA3OX2*, and *TGA1*. We also found that there was not a strict one-to-one correspondence between miRNAs and their target genes. Sometimes, there may be one miRNA targeting multiple genes, which also adds to the complexity of its regulatory network.

### 2.4. GO Analysis of Differentially Expressed Target Genes of miRNAs

To further explore the function of miRNAs target genes that are differentially expressed in WT and *eno2^−^* mutant, we used *Arabidopsis* GO (Gene Ontology) annotation database (http://www.geneontology.org/ (accessed on 30 April 2021)). GO annotation was performed on the target genes of miRNAs in WT and *eno2^−^* seedlings which were grown for 7 days (Figure 6). According to statistics, 4683 differential miRNA target genes were annotated to 45 GO terms, among which 1963 differential miRNA target genes were annotated to the biological process, 873 differential miRNA target genes were annotated to the molecular function, and 1847 differential miRNA target genes were annotated to the cellular component.

In 21 biological processes, the DEM target genes in WT and *eno2^−^* are mainly enriched in cellular process, metabolic process, biological regulation, regulation of the biological process, response to the stimulus, developmental process, and other GO terms. Among the nine molecular functions, the DEM target genes in WT and *eno2^−^* were mainly concentrated in binding, catalytic activity, and transporter activity. In the 13 cell components, the DEM target genes in WT and *eno2^−^* are mainly enriched in the cell part, cell, organelle, membrane, and membrane part, and other GO terms.

### 2.5. KEGG Pathways Analysis of Differentially Expressed Target Genes of miRNAs

To know more deeply and intuitively the metabolic pathways enriched by target genes that DEMs in WT and *eno2^−^*, and then explain the molecular mechanism that *AtENO2* gene affects miRNAs and seed germination, KEGG (Kyoto Encyclopedia of Genes and Genomes) analysis (https://www.genome.jp/kegg/pathway.html (accessed on 30 April 2021)) was performed on the target genes of DEMs screened. Additionally, the top 20 pathways were selected according to their enrichment levels (Figure 7).

According to the analysis, 133 target genes of miRNAs were enriched in 69 metabolic pathways. Among the 20 top enrichment pathways, the protein processing in the endoplasmic reticulum had the most abundant miRNA target genes (up to 17), followed by the spliceosome (13), and plant hormone signal transduction (13). All of them were significantly enriched (q-value < 0.05). Additionally, mRNA surveillance pathway, terpenoid backbone biosynthesis, fatty acid degradation, ribosome biogenesis in eukaryotes, and tryptophan metabolism were also significantly enriched.

### 2.6. qRT-PCR Validation of miRNAs and Target Genes

With the analysis of KEGG and GO in DEMs of WT and *eno2^−^*, we finally found the target genes of these 15 miRNAs (ath-miR167a-5p, ath-miR167b, miR8165-z, ath-miR393a-5p, ath-miR393b-5p, miR9736-z, miR10186-z, ath-miR866-3p, miR163-z, miR5059-z, ath-miR5665, ath-miR857, ath-miR399b, ath-miR399c-3p, and miR399-y) were related to plant hormones. We performed qRT-PCR to validate the deep sequencing results with selected 15 miRNAs. For this purpose, we used the WT seedlings growing for seven days as a control to figure out the expression levels of *eno2^−^* seedlings growing in the same conditions with three technical and three biological replicates. Six miRNAs (ath-miR393a-5p, ath-miR393b-5p, ath-miR866-3p, miR163-z, ath-miR5665, and ath-miR857) were up-regulated in qRT-PCR analysis showing a positive correlation with deep sequencing results. Similarly, the other nine miRNAs were down-regulated in both qRT-PCR and high-throughput sequencing results (Figure 8). We also validated the 18 corresponding target genes (*ARF6*, *ARF8*, *ARF5*, *ARF2*, *ARF3*, *GA3OX2*, *TGA1*, *TSN2*, *ATGSL10*, *RR19*, *INVC*, *TIR1*, *DPBF3*, *ARR16*, *ACAT2*, *CTR1*, *MED37E,* and *PXMT1*) which related to seed germination. Five target genes (*ARF5*, *ARF2*, *RR19*, *TIR1,* and *PXMT1*) were down-regulated in qRT-PCR analysis, and the other 13 target genes were up-regulated. The expression level of miRNAs and target genes showed the opposite trend (Figure 9).

## 3. Discussion

Seed germination is a critical stage of normal plant growth and development. External environmental factors and internal genetic factors play a decisive role in the process from dormancy to germination, in which plant hormones play an important role in the process of seed germination. Many studies have shown that different miRNAs can regulate seed germination by targeting key genes in the phytohormone signal transduction pathway. In our study, the germinability in *eno2^−^* was higher than that of WT. After deep sequencing, we analyzed the target genes of key miRNAs, the GO and KEGG pathway analysis of target genes. We found that the pathway of plant hormone signal transduction pathway was significantly enriched, including 15 miRNAs and 18 target genes involved in seed germination (Appendix A). In the following, we will discuss the role of these miRNAs and their corresponding target genes in phytohormone signal transduction and seed germination.

### 3.1. Gibberellin and Abscisic Acid

Gibberellin (GA) and abscisic acid (ABA) are the two main factors that determine seed maturity, dormancy, and germination [37,38]. Previous researches showed that GA and ABA are antagonistic to each other in the regulation of seed dormancy and germination [39]. ABA induces seed dormancy and inhibits seed germination, while GA can relieve seed dormancy and promote seed germination [40,41,42]. In the present study, the GA-related miR10186, ath-miR399, and miRNA8165 were differentially expressed (Appendix A), whose target genes were *GA3OX2*, *INVC*, and *AtTudor2.* There was a negative regulatory relationship between miRNAs and target genes. The expressions of miR10186, ath-miR399, and miRNA8165 were down-regulated, while the expressions of *GA3OX2*, *INVC*, and *TSN2* were up-regulated, which was consistent with the results of qRT-PCR (Figure 6). The *GA3OX2* gene encodes a key enzyme for GA synthesis, which catalyzes the conversion of inactive GA into biologically active GA. During seed germination, *AtGA3OX2* was regulated by *LEC2* and *FUS3* to synthesize GA to promote seed germination [43]. *INVC* belongs to the glycoside hydrolase 100 family, and the *Arabidopsis invc* resulted in repressing seed germination. However, exogenous application of GA could save the repression, and it was speculated that *INVC* could activate GA signaling pathway to promote seed germination [44]. The TSN2, a protein with SN-Tudor domains, is involved in the control of animal cell growth and development. It was reported that the expression of *AtGA20OX3*, a key enzyme of GA biosynthesis, was significantly down-regulated in the *Arabidopsis tsn2* mutation, and the seed germination of the mutant was delayed, which indicated that *TSN2* might regulate seed germination by regulating the biosynthesis of GA [45]. In the present study, the up-regulated expression of *GA3OX2*, *INVC*, and *TSN2* could increase the GA content in *eno2^−^* seeds and promoted seed germination.

In the present study, we identified miR5059 involved in regulating the ABA signaling pathway (Appendix A). It was found that there were two target genes, *ACAT2* and *DPBF3,* through target gene prediction analysis. *ACAT2* encodes acetoacetyl-CoA thiolase, and the acetoacetyl CoA precursor derived from *ACAT2* is the basis for the synthesis of isoprenoids required for normal plant growth and development. ABA is a sesquiterpene substance composed of isoprene. *ACAT2* indirectly regulated seed germination by affecting ABA synthesis [46]. *DPBF3* belongs to the family of *bZIP* transcription factors, and encodes ABA response element-binding proteins, which are involved in regulating seed development and germination [47]. Using RNAi technology to reduce the expression of *DPBF3* gene in *Arabidopsis thaliana*, there was no change in seed development, but the germination was advanced [48]. In our study, the expression of miR5059 was down-regulated, and the expression of *ACAT2* and *DPBF3* were up-regulated, which reduced the inhibitory effect of ABA in *eno2^-^* on seed germination. In brief, these results indicated that the down-regulated miR10186, ath-miR399, and miR5059 participated in the regulation of seed germination through the ABA and GA signaling pathways, and promoted *eno2^−^* seed germination.

### 3.2. Auxin

As one of the earliest plant hormones used in plant science research, auxin plays a unique role in regulating directional growth, tissue differentiation, organogenesis, morphogenesis, apical dominance, and flowering. In recent years, it has been found that auxin can also involve in the germination and dormancy of plant seeds [49,50,51,52]. Usually, a low concentration of auxin can promote seed germination, while a high concentration can inhibit seed germination. In transgenic *Arabidopsis thaliana iaaM-OX*, it can produce a large amount of auxin, and the corresponding seed germination was seriously inhibited [53], which indicated that auxin can play a role in the process of seed germination, and mainly showed an inhibitory effect. *Auxin response factor* (*ARF*) gene family is a new family of transcription factors, which can inhibit or activate auxin response genes by specifically binding to the TAGTCTC sequence in the promoter of auxin response genes.

In our results, there are ath-miR167a-5p, ath-miR167b, ath-miR5665, ath-miR866-3p, and miR10186-z targeting *ARF6, ARF8, ARF5, ARF 2,* and *ARF3* (Appendix A), respectively. Analysis of Appendix A showed that the expression of ath-miR167a-5p, ath-miR167b, and miR10186-z were down-regulated in *eno2^-^*, while the expression of ath-miR5665 and ath-miR866-3p was up-regulated. It was known that *ARF6/8* targeted by miR167 acted as a positive regulator during adventitious root growth in *Arabidopsis*. Besides, *ARF6/8* can inhibit stamen elongation and flower maturation [54]. *ARF5* encodes a transcription factor (IAA24) that mediates the formation of hypocotyls. Similar to *ARF1*, *ARF5* binds to the auxin-responsive elements (AREs) [55]. In *ARF2* mutants, there were many defects, including the enlargement of rosette leaves, the decrease of fertility, and the elongation of the hypocotyl [56]. TIR1, as an auxin receptor, mediates auxin-regulated transcription [57]. In our results, ath-miR393a-5p targeted *TIR1*, and up-regulated in *eno2^−^* (Appendix A). Based on the changes of miRNAs expression and its negative regulation on the corresponding target genes, it was speculated that the auxin content in *eno2^−^* decreased, which weakened seed dormancy and promoted seed germination.

### 3.3. Cytokinin

*Arabidopsis response regulator* (*ARR*) is an important part of CK signal transduction. Among them, *ARR4*, *ARR5,* and *ARR6*, which have been reported, can negatively regulate *ABI5* expression and make CK resist the inhibition of ABA. *ABI4* can directly combine with the promoter to inhibit the transcription of *ARR6*, *ARR7*, and *ARR15* [58,59]. In our experimental results, it was found that ath-miR857 and miR9736-z target *RR19*, and *ARR16* (Appendix A), respectively. Both *RR19* and *ARR16* are known to play an indispensable role in responding to cytokinin signal transduction pathways. Compared with the results of deep sequencing of WT and *eno2^−^*, it was found that ath-miR857 was up-regulated, while miR9736-z was significantly down-regulated in *eno2^−^* (Appendix A). Combined with phenotypic analysis, it can be inferred that miR9736-z is more significant in targeting *ARR16* in this process.

### 3.4. Salicylic Acid

Salicylic acid (SA), as a signal molecular, is involved in regulating various physiological functions in plants [60]. At present, it has been found that exogenous spraying of SA in eggplants [61], carrots [62], and sunflowers [63] can promote seed germination. Through target genes prediction, it was found that, in addition to *ARF2* and *GA30X2*, target genes of miR10186 also include *TGA1* related to the salicylic acid signaling pathway. *TGA1* can positively regulate SA synthesis [64]. In the present study, the up-regulated expression of *TGA1* promoted SA synthesis. The increase of SA content in *eno2^−^* can promote seed germination.

### 3.5. Others

Besides hormonal pathway, miRNAs also regulate seed germination through other signaling pathways. During seed germination, miR163 and its target gene *PXMT1* were mainly expressed in radicles. The miR163 mutants or *PXMT1* overexpression lines showed delayed seed germination under continuous light conditions, indicating that miR163 and target gene *PXMT1* promoted seed germination during the early development in seedlings [65]. In our study, the expression of miR163 was significantly up-regulated, which promoted *eno2^−^* seed germination. In addition to the *AtTudor2* gene, the target genes of miR8165 included *ATGSL10*. *ATGSL10* encodes GSL10, a member of the glucan synthase (GSL) family, which is essential for male gametophyte development and plant growth, and plays a role in microspore entry into mitosis. The *atgsl10* resulted in disruption of microspore division symmetry, and delayed seed germination [66]. In our study, the expression of *GSL10* in *eno2^−^* was up-regulated, which promoted seed germination.

## 4. Materials and Methods

### 4.1. Plant Materials and Growth Conditions

The *Arabidopsis thaliana* WT was the Columbia (Col-0) ecotype background. The T-DNA insertion mutant *eno2^−^* (SALK_021737) was acquired from *Arabidopsis* Biological Resource Center (https://abrc.osu.edu/stocks/number/SALK_021737 (accessed on 30 April 2021)). Seeds were surface-sterilized for 7 min in 2% sodium hypochlorite (Lvtao Environmental Technology, Cangzhou, China) and rinsed five times with sterile water. Then the seeds were cold-treated at 4 °C in the dark for 3 days. The seeds germinated on half-strength MS (1/2 MS) (Coolaber, Beijing, China) medium supplemented with 0.05% MES buffer (Coolaber, Beijing, China), 1.2% sucrose (w/v) (Coolaber, Beijing, China), and 0.8% agar (*w/v*) (Coolaber, Beijing, China). The seeds were cultivated at 21 ± 2 °C under 16 h light/8 h dark.

### 4.2. Seed Germination Rate Analyses

The seeds were cultivated on 1/2 MS medium and the seed germination rates were counted the next day after planting. Seed germination rates were counted at 10 o’clock every morning for one week. Germinability refers to the percentage of seeds that can germinate normally after 3 days of growth to the total number of seeds. The average seed germination rate was obtained from three experimental replicates, and about 30 seeds were observed for each genotype. The seed germination rate was obtained by counting the number of germinated seeds divided by the total number of seeds tested. After growth for 7 days, the whole seedlings were collected and immediately frozen in liquid nitrogen. The samples are stored at −80 °C until RNA extraction.

### 4.3. Small RNA Library Construction and Sequencing

Total RNAs were extracted from each sample using TRIzol reagent (Invitrogen, Carlsbad, CA, USA); the RNA molecules in a size range of 18–30 nt were enriched by polyacrylamide gel electrophoresis (PAGE). Then the small RNA of the recovered 18–30 nt was attached to the 3′ end and 5′ end adapter, and performs reverse transcription and PCR amplification on the small RNA connected with the adapters. Finally, the 140 bp band was recovered and purified by agarose gel electrophoresis to complete the library construction. The constructed library used Agilent 2100 Bioanalyzer (Palo Alto, CA, USA) and ABI StepOnePlus Real-Time PCR System (Applied Biosystems, Foster City, CA, USA) for quality and yield detection, and sequencing using the Illumina NovaSeq6000 by Gene Denovo Biotechnology Co. (Guangzhou, China). Additionally, the single-end sequencing was used in our experiment. Each sample produces 10 M reads. We submitted our raw data to the SRA database (https://submit.ncbi.nlm.nih.gov/subs/sra/ (accessed on 30 April 2021)), and the submission number was SUB9500128.

### 4.4. Bioinformatic Analysis

The raw reads were first cleaned up, including eliminating low-quality reads, removing adapter sequences, and filtering reads containing poly A. Then the length distribution and base distribution of clean reads were analyzed, and 16–30 nt length small RNAs were selected as sRNA for subsequent analysis. The software Blastall 2.2.25 (BLASTN) was used to compare with the rRNA, scRNA, snoRNA, snRNA, and tRNA in GenBank (Release 209.0, https://www.ncbi.nlm.nih.gov/genbank/ (accessed on 30 April 2021)) database, and the tags with an identity more than 97% were removed. Meanwhile, the software Blastall 2.2.25 (BLASTN) was used to compare with the rRNA, scRNA, snoRNA, snRNA, and tRNA in Rfam (Release 11.0, http://rfam.xfam.org/ (accessed on 30 April 2021)) database, and the tags with an identity more than 97% were removed [67,68]. All of the clean reads were also aligned with the reference genome (http://plants.ensembl.org/Arabidopsis_thaliana/Info/Index (accessed on 30 April 2021)). We used Bowtie (version 1.1.2) software to align the tags to the genome to determine the specific location of the tags from the genome. According to the results of the alignment to the genome, as well as the position of exons and introns in the genome, we can find the tag sequences from mRNA degradation fragments and remove these tag fragments. At the same time, the tags of repetitive sequences aligned in the genome were removed. All of the clean reads were then searched against the miRBase database (Release 22, http://www.mirbase.org/search.shtml (accessed on 30 April 2021)) to identify existing (species studied) miRNAs. We used Bowtie (version 1.1.2) software to compare the remaining clean tags with the miRNA sequences of *Arabidopsis thaliana* in miRBase, and obtained the content and base distribution of the exist miRNA in the sample. So far, the miRNA sequences of some species were still not included in the miRBase database. For those species, the miRNAs alignment with other species was a dependable way to identify the conserved miRNAs. All of the unannotated reads were aligned with the reference genome. According to their genome positions and hairpin structures predicted by software mirDeep2 (https://www.osc.edu/book/export/html/4389 (accessed on 30 April 2021)), the novel miRNA candidates were identified [69]. All identified miRNAs were analyzed quantitatively, with target gene prediction, and target gene functional annotation. The miRNA expression levels were calculated and normalized to per million transcripts (TPM) based on miRNA expression in each sample. The formula is as follows: TPM = actual miRNA counts/total counts of clean tags ×10^6^ [70]. Meanwhile, the fold change values were calculated in WT and *eno2^−^*, and the DEMs were screened by edgeR according to the fold change values. We identified DEMs with a fold change ≥ 2 and *p*-value ≤ 0.05 as significant DEMs. According to the principle of target genes prediction, target genes of the total miRNAs were predicted by the software PatMatch (Version 1.2). The predicted target gene sequences of miRNAs were functionally annotated with GO (http://www.geneontology.org/ (accessed on 30 April 2021)) and KEGG (https://www.genome.jp/kegg/pathway.html (accessed on 30 April 2021)) databases to obtain functional information of target genes in biological processes, cell components, and molecular functions.

### 4.5. Validation of miRNA and Target Gene Expressions with qRT-PCR Analysis

RNA was extracted from seedlings grown for one week to verify the miRNA sequencing results by using Eastep^®^ Super Total RNA Extraction Kit (Promega, Madison, WI, USA). The miRNA was detected by tailing reaction RT-PCR using miRNA First Strand cDNA Synthesis kit (Sangong Biotech, Shanghai, China). A total of 20 μL RT reaction contained 10 μL 2 × miRNA RT Solution mix, 2 μL miRNA RT Enzyme mix, 10 μg total RNA, and RNase-free water were used to generate single-stranded cDNA for miRNAs. After mixing gently, we performed RT reaction at 60 min at 37 °C, 5 min at 85 °C, and then holding at 4 °C. The obtained cDNA reaction solution was diluted 50 times and then used as the template for qRT-PCR; 15 miRNAs related to seed germination (ath-miR167a-5p, ath-miR167b, miR8165-z, ath-miR393a-5p, ath-miR393b-5p, miR9736-z, miR10186-z, ath-miR866-3p, miR163-z, miR5059-z, ath-miR5665, ath-miR857, ath-miR399b, ath-miR399c-3p, and miR399-y) were selected to validate miRNA expression of deep sequencing. Besides, we also validated the 18 corresponding target genes (*ARF6*, *ARF8*, *ARF5*, *ARF2*, *ARF3*, *GA3OX2*, *TGA1*, *TSN2*, *ATGSL10*, *RR19*, *INVC*, *TIR1*, *DPBF3*, *ARR16*, *ACAT2*, *CTR1*, *MED37E,* and *PXMT1*) which related to seed germination. The specific forward primers and universal reverse primers were designed by Primer 5.0 (Primer Software, Premier, QC, Canada) (Appendix A). The qRT-PCR was performed using TransStart^®^ Top Green qPCR SuperMix (TransGen, Beijing, China) on the ABI 7500 Real-Time PCR system (Applied Biosystems, Foster City, CA, USA). The 20 μL reactions contained 2 μL reverse transcription cDNA products, 10 μL qPCR SuperMix, 2 μL primer mix and supplemented with nuclease-free water to 20 μL. The qRT-PCR reactions were performed as follows: 94 °C for 30 s, followed by 40 cycles of 94 °C for 5 s, and 60 °C for 30 s. Each reaction has three technical replicates, and the expression levels were calculated using the 2^−ΔΔCt^ method [71]. *UBQ5* gene and *U6* snRNA were used as the internal standard, and Student’s t-test was used to analyze the miRNA expression between WT and *eno2^−^* seedling with the GraphPad Prism 8.0 (GraphPad Software, La Jolla, CA, USA).

## 5. Conclusions

At present, hundreds of genes have been found to be related to seed germination. Our research showed that *AtENO2* was involved in the regulation of seed germination. After *AtENO2* mutation, seed germination was advanced. Previous studies have shown that miRNAs played a pivotal role in seed germination. In our research, we found that various miRNAs were differentially expressed in *eno2^−^* through deep sequencing of small RNA, including miR10186, ath-miR399, miRNA8165, and ath-miR393a-5p, and corresponding target genes regulated seed germination by regulating the synthesis of GA, auxin, and other plant hormones. These findings lay a foundation for further exploration of the molecular mechanism of *AtENO2* regulating seed germination.

## Figures and Tables

**Figure 1 ijms-22-05088-f001:**
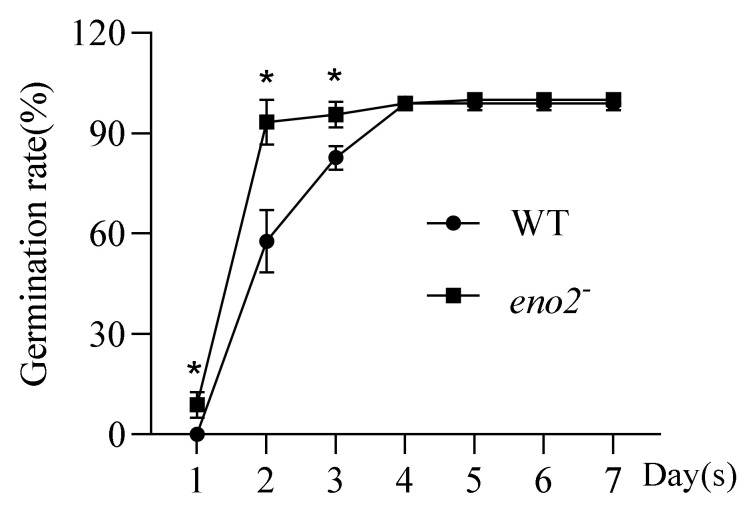
Statistical analysis on germination of WT and *eno2^−^.* The X-axis represented the growth days, and the Y-axis indicated the germination rate. Three independent biologicals and three technical replicates were employed; the error bars showed the SD (standard deviations). The significant difference was compared using the Student’s t-test. * *p* ≤ 0.05.

**Figure 2 ijms-22-05088-f002:**
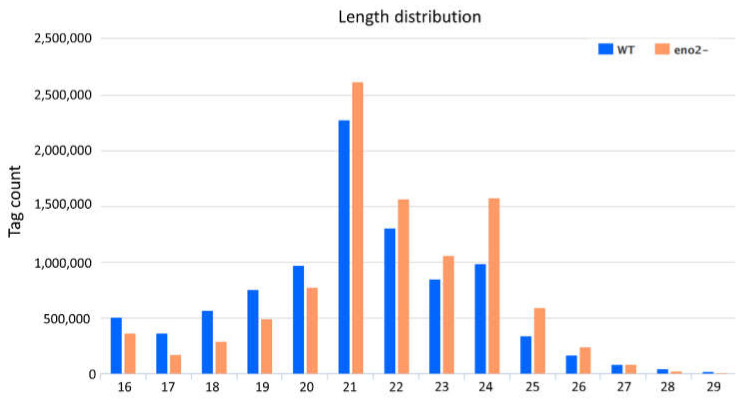
Length distribution of small RNAs in WT and *eno2^−^.* The X-axis represented the length of small RNAs, and the Y-axis meant the tag counts. The blue bar charts indicated WT, and the orange bar charts showed the *eno2^−^.*

**Figure 3 ijms-22-05088-f003:**
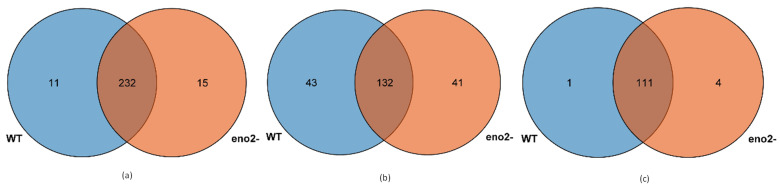
Numbers of miRNAs in WT and *eno2^−^*. (**a**) Existing miRNAs; (**b**) Conserved miRNAs; (**c**) Novel miRNAs.

**Figure 4 ijms-22-05088-f004:**
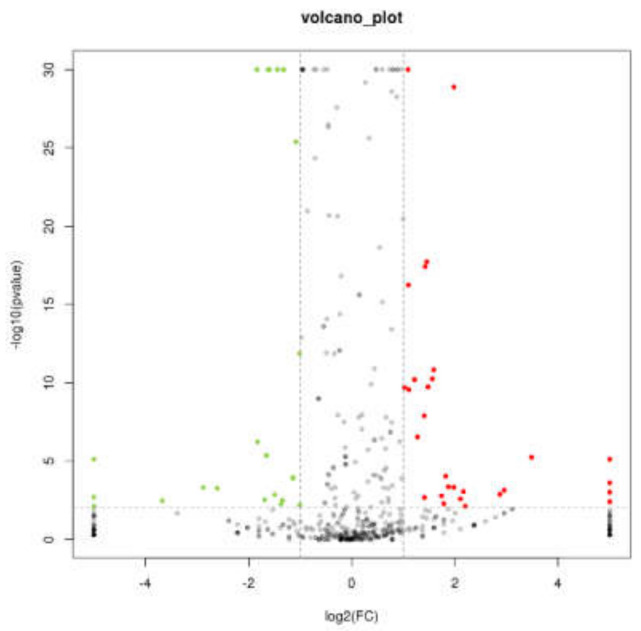
The volcano map of DEMs in WT and *eno2^−^.* The X-axis represented the log value of the difference multiple between WT and *eno−^-^*, and the Y-axis showed the negative Log10 value of the FDR of the difference between WT and *eno2^−^*. The red dots meant up-regulated miRNAs, while the green represented down-regulated miRNAs. The gray dots were no significant difference. FDR < 0.05; log2 fold change ≥2.

**Figure 5 ijms-22-05088-f005:**
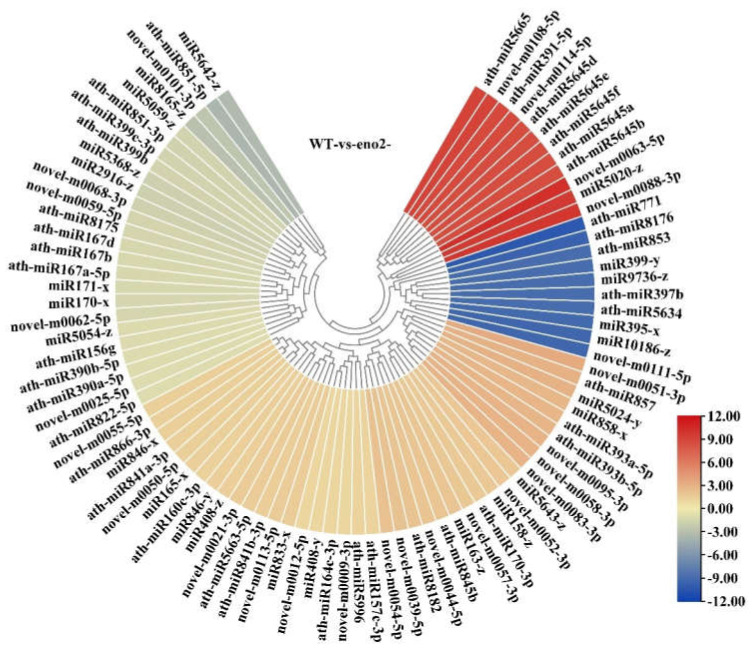
The heatmap of significantly DEMs in WT and *eno2**^−^*. The heatmap was drawn with log2 FC by using TBtools [36]. The red to yellow colors represented significantly up-regulated expression miRNAs, and the yellow to blue colors represented significantly down-regulated expression miRNAs.

**Figure 6 ijms-22-05088-f006:**
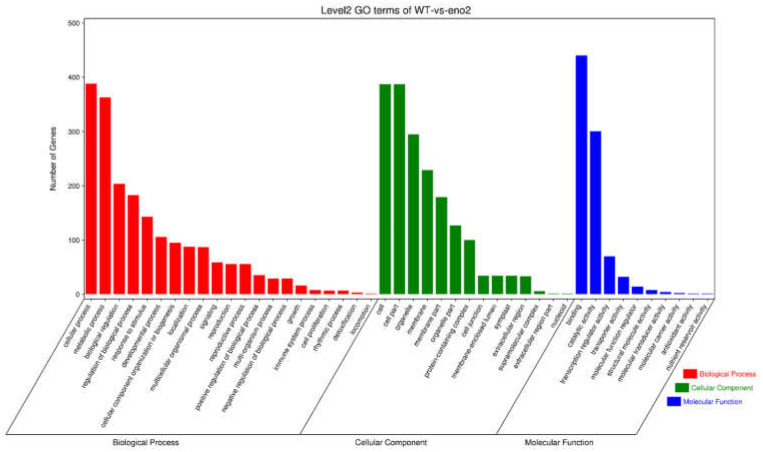
GO term classification for DEMs target genes in WT and *eno2^−^*. The *X*-axis showed GO terms, and *Y*-axis meant the number of genes. The red bar charts represented the biological process, the green bar charts indicated the molecular function and the blue bar charts represented the cellular component.

**Figure 7 ijms-22-05088-f007:**
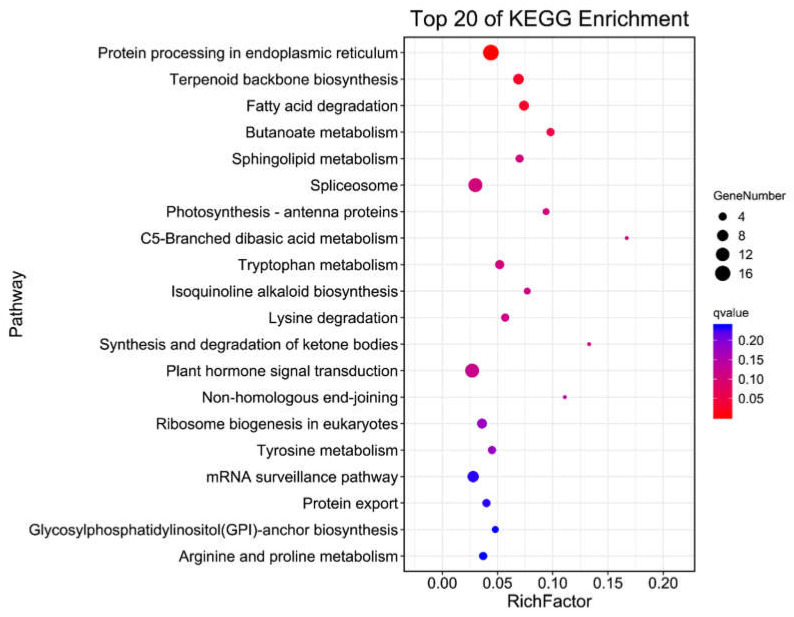
KEGG pathways analysis of DEMs target genes in WT and *eno2^−^*(TOP 20). Q-value represented enrichment degree, blue to red colors indicated decreasing values. The smaller the q-value, the more significant the enrichment. The size of the circle showed the amount of enrichment, and the bigger the circle size, the more genes enriched in the related pathways.

**Figure 8 ijms-22-05088-f008:**
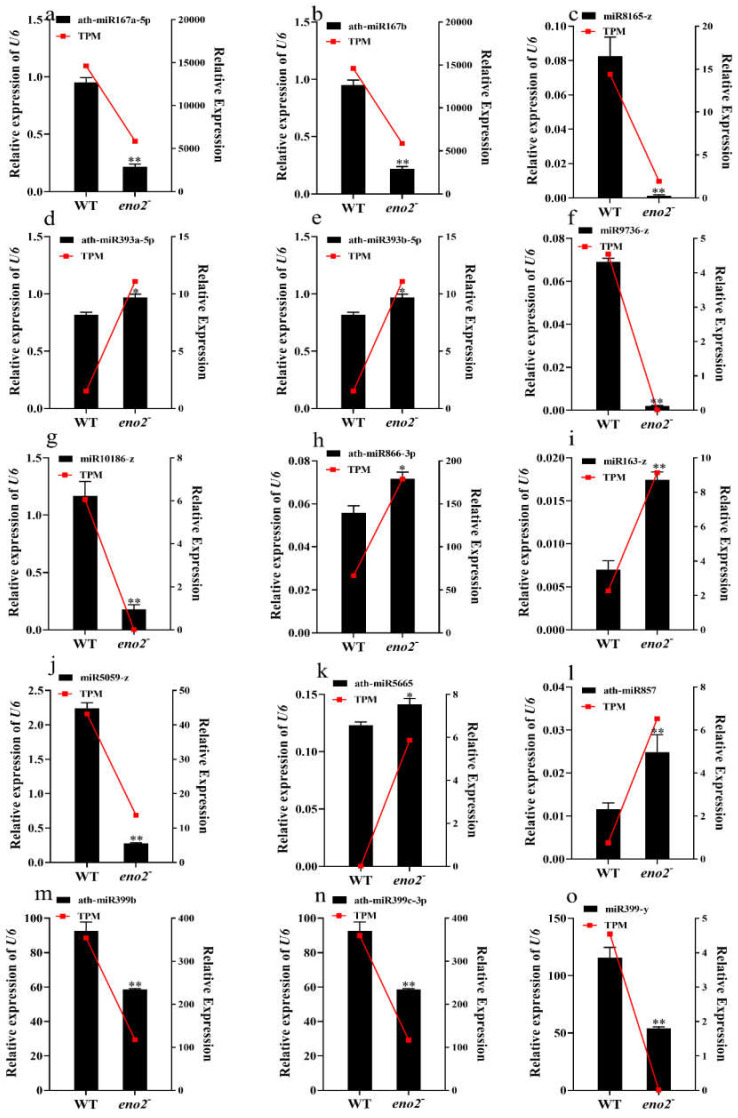
qRT-PCR validation of 15 miRNAs: (**a**–**o**) Represented different miRNAs expression levels with deep sequencing and qRT-PCR. The TPM value was taken as the relative expression level of deep sequencing miRNAs, which was shown as the red line, corresponding to the Y-axis on the right side. The qRT-PCR validation used the black bar chart, corresponding to the Y-axis on the left side, and *U6* was used for each sample as an endogenous control. Three independent biologicals and three technical replicates were employed; the error bars show the SD (standard deviations). The significant difference was compared using the Student’s t-test. * *p* ≤0.05; ** *p* ≤ 0.01.

**Figure 9 ijms-22-05088-f009:**
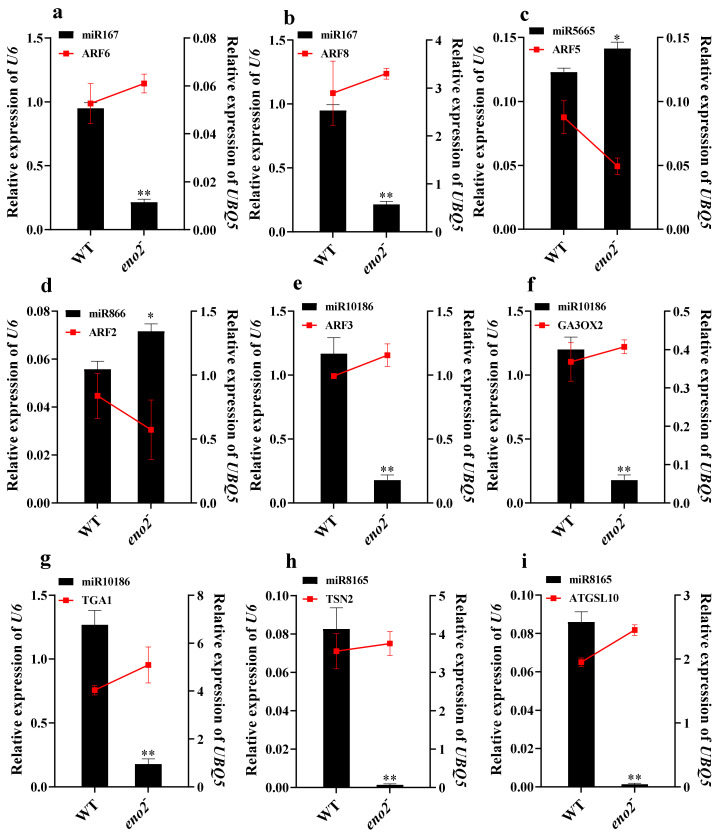
qRT-PCR analyses of the relative expression levels of various predicted target genes: (**a**–**r**) Represented different target gene expression levels with qRT-PCR. The expression level of miRNAs with the black bar chart, corresponding to the Y-axis on the left side, and *U6* was used for each sample as an endogenous control. The predicted target genes’ expression level with the red line chart, corresponding to the Y-axis on the right side, and *UBQ5* as the internal standard. Three independent biologicals and three technical replicates were employed; the error bars show the SD. The significant difference was compared using the Student’s t-test. * *p*≤ 0.05; ** *p* ≤ 0.01.

**Table 1 ijms-22-05088-t001:** The tags’ type and abundance of WT and *eno2^−^.*

	Unique (WT)	Abundance (WT)	Unique (*eno2^−^*)	Abundance (*eno2^−^*)
**rRNA**	66,175 (6.28%)	2,068,165 (21.41%)	58,311 (5.73%)	2,196,583 (21.81%)
**scRNA**	741 (0.07%)	5638 (0.06%)	784 (0.08%)	8955 (0.09%)
**snRNA**	20,029 (1.90%)	1,533,088 (15.87%)	19,108 (1.88%)	1,306,885 (12.98%)
**snoRNA**	2916 (0.28%)	18,664 (0.19%)	3127 (0.31%)	24,790 (0.25%)
**tRNA**	4756 (0.45%)	241,070 (2.50%)	4960 (0.49%)	350,910 (3.48%)
**exon_sense**	73,972 (7.02%)	587,871 (6.08%)	61,192 (6.02%)	528,694 (5.25%)
**miRNA**	5407 (0.51%)	1,320,827 (13.68%)	5629 (0.56%)	1,532,594 (15.22%)
**unann**	714,821 (67.84%)	2,733,234 (28.29%)	662,205 (65.11%)	2,632,778 (26.14%)
**total**	1,053,660	9,661,299	1,017,006	10,071,178

rRNA—ribosomal RNA; scRNA—small cytoplasmic RNA; snRNA—small nuclear RNA; snoRNA—small nucleolar RNA; tRNA—transfer RNA; miRNA—microRNA; unann—unannotated RNA.

## Data Availability

The data are available upon reasonable request from qualified researchers.

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
