# Peer review of "Deep Sequencing of Small RNA Reveals the Molecular Regulatory Network of *AtENO2* Regulating Seed Germination"

_ijms, 2021, doi:10.3390/ijms22105088_

Round 1
Reviewer 1 Report
This manuscript entitled " Deep sequencing of small RNA reveals the molecular regulatory network of ENO2 regulating seed germination" is a comprehensive study by Wu et al, investigating the small RNA regulations between wild type and ENO2 mutant of Arabidopsis. I have the following comments:
- Abstract: no mention of the species name of the plant
- Line 140-143 ...."which conforms to the laws of biology" What is the laws of biology here? And i do not agree "the number of 16nt length distributions is a little too large " can the authors explain a bit more? From glancing the Figure 2, it is more obvious that the 24nt peak of the mutant is significantly higher.
- Table 1 - the unann (unannotated) small RNAs are quite high , can the authors do a quick check of what are these? Environmental contaminations?Bacterial reads?
- Line 193- fold change> 2 . Is it absolute fold change or log2 fold change?
- Line 236 - What is the criteria for selecting the 15 miRNAs for validations?
- Materials and methods: line 411 "connect the 3' joint ..." did the authors mean ligate the 3' end and 5'end? It is advised the authors to proofread the whole manuscript.
- Line 416: HiSeq model is missing. Paired-end or Single-end sequencing? Sequencing depth?
- Bioinformatics analysis: software/pipeline information are missing. For example, what did the authors used to align with reference genomes? Please check the entire methods section and add missing information.
Author Response
This manuscript entitled " Deep sequencing of small RNA reveals the molecular regulatory network of ENO2 regulating seed germination" is a comprehensive study by Wu et al, investigating the small RNA regulations between wild type and ENO2 mutant of Arabidopsis. I have the following comments:
- Abstract: no mention of the species name of the plant
Response:
Thanks for your suggestion, we have added the species name of the plant in the abstract, which is also marked red (line 13).
- Line 140-143 ...."which conforms to the laws of biology" What is the laws of biology here? And i do not agree "the number of 16nt length distributions is a little too large " can the authors explain a bit more? From glancing the Figure 2, it is more obvious that the 24nt peak of the mutant is significantly higher.
Response:
Thanks for your revision, we have revised our inappropriate statement. The modified section has been marked red (line 134-137).
- Table 1 - the unann (unannotated) small RNAs are quite high, can the authors do a quick check of what are these? Environmental contaminations? Bacterial reads?
Response:
The tags used to analyze small RNAs have been removed for tags with a length less than 18 nt or an abundance less than 2. This part of the tags is not used for subsequent small RNA identification or tag type identification. These data were directly attributed to the unann (unannotated), resulting in the proportion of this part of data is too large, and did not belong to environmental contaminations or bacterial reads. We counted the number of tags with a length less than 18 nt or an abundance less than 2 in this part of the reads, and counted the tag types of unann in the process, and it could be found that more reads were tags with length less than 18 nt or abundance less than 2. Specific data were shown in the following table.
|
sample |
total_unann |
samll_than_18nt |
only_one_count |
final_unann |
|
WT |
719765 |
135928 (18.89%) |
447704 (62.20%) |
136133 (18.91%) |
|
eno2 |
664113 |
75666 (11.39%) |
420947 (63.38%) |
167500 (25.22%) |
- Line 193- fold change> 2. Is it absolute fold change or log2 fold change?
Response:
It is log2 fold change, we have corrected it and marked red (line 185-189).
- Line 236 - What is the criteria for selecting the 15 miRNAs for validations?
Response:
With the analysis of KEGG and GO in significant differential miRNA of WT and eno2-, we finally found the target genes of these 15 miRNAs were related to plant hormone. Because the phytohormone has a close relationship with seed germination, so we chose these 15 miRNAs for further validation. We have also added the reasons and marked them in red (line 236-240). Thank you for reminding us.
- Materials and methods: line 411 "connect the 3' joint ..." did the authors mean ligate the 3' end and 5'end? It is advised the authors to proofread the whole manuscript.
Response:
We mean that the small RNA of the recovered 18-30nt was attached to the 3' end and 5' end adapter for the further PCR experiment. And we have checked the whole manuscript (line 411-414).
- Line 416: HiSeq model is missing. Paired-end or Single-end sequencing? Sequencing depth?
Response:
We used Illumina NovaSeq6000 by Gene Denovo Biotechnology Co, and the single-end sequencing was used in our experiment. Each sample produces 10M reads (line 416-420).
- Bioinformatics analysis: software/pipeline information are missing. For example, what did the authors used to align with reference genomes? Please check the entire methods section and add missing information.
Response:
We used Bowtie (version 1.1.2) software to align the tags to the genome to determine the specific location of the tags from the genome. According to the results of alignment to the genome, as well as the position of exons and introns in the genome, we can find the tag sequences from mRNA degradation fragments and remove these tag fragments. At the same time, the tags of repetitive sequences aligned in the genome were removed. We have added the details of the software information and the methods section. All these corrections are marked red (line 425-442; line 455-459).

Reviewer 2 Report
The paper by Wu et al. aimed to identify microRNA in wild type and eno2-, a mutant for a key gene involved in glycolysis and other pathways such as those controlling plant growth and abiotic stress responses. In their work, the authors first sequenced small RNA libraries and then they identified potential mRNA targets. Finally, fifteen miRNAs along with their targets were chosen to validate their expression by qRT-PCR.
Although the paper showed interesting results, I found several major concerns that should be addressed:
In my point of view, one of the main concerns is the number of sequenced samples. Indeed, at line 131 the authors reported that only two libraries were produced (one for wild type and one for the mutant). It is not clear to me how the authors can identify differentially expressed miRNAs by using only a single replicate rather than three biological samples that are usually required to avoid spurious results. Which kind of statistics the authors applied to identify differentially expressed miRNA? Parametric tests such as DEseq2?
Another major concern regards the lack of rigor. For example, in Figure 1 is reported the statistical analysis of WT and mutant. However, in the same figure the construct of ENO2- is also shown. I suggest moving it in supplementary material or delete it if not necessary. In addition, in the same figure, I see the standard deviation but it is not clear if the difference between WT and ENO2- in terms of germination rate is statistically different (no p.values are reported). Which kind of statistical test was performed with these data? Please add also the legend for X-axis. Another major concern regards inconsistencies between the results, text and figures. For example, in line 104 the authors reported that 590 miRNAs were differentially expressed but, in the results section, (line 166) they write 591. In addition, in line 155 they reported that:” was 243 and 247 and 232 are in common”. Figure 3 showed different numbers. This is not exhaustive, since other numbers did not match between figures and text.
In my point of view table 1 is confounding. What is the difference between exist_miRNA and exist_miRNA_edit? In addition, it is not clear to me what the authors refer to with tags. Do you mean mapped reads that have a match with microRNA? The legend of table 1 is not sufficient. The authors should provide more details to get it self-explaining.
In line 149 the authors reported as follows: “Statistical analysis showed that in WT and eno2-, the number of tag species was 1053660 and 1017006, and the tag abundance was 9661299 and 10071178, respectively (Tab. 1).” Where is the statistical results? Which kind of statistical test the authors applied?
Minor concerns:
I suggest using coma or point when writing numbers in the text
Line 132. I think these numbers refer to row reads and not to clean reads.
The authors should be more rigorous in writing gene names. The gene names are not always in Italics (i.e. see line 184, 246-247, 79, 95). This is not exhaustive
Line 137-138. “but the expression trend of small RNAs with different lengths in the library is similar”. Where we can see this data? If they are not reported I would suggest adding (i.e. data not reported) or remove it if not indispensable
Figures 10 and 11 should be moved in supplementary.
In line 147-148 of my pdf, strange symbols appeared. What is rRNAetc> ?
Lines 168-171. I think is more appropriate moving this part in the material and methods section or in the legend
Line 173. What the authors mean when they say: “we mapped”? I think this word is not appropriate
Figure 5. Which kind of data the authors used to drawn this plot? Legend is not sufficient, and it is not clear to me if the authors used TPM data or transformed one.
Line 224:” According to the analysis, 133 target genes of miRNA were enriched in 69 metabolic pathways. Among the 20 enrichment pathways …”. Are the metabolic pathways 69 or 20? Did the authors mean among the 20 top enriched pathways?
Figure 8 and 9. Is it possible to apply a statistical test? This could help the readers to understand if the difference in such miRNA expression in WT and eno2- is significant or not.
In the material and methods section many details regarding the company and their city where the reagents were bought are missing. Please add more details in the material and methods section.
Line 423. How the authors aligned the clean reads to the Genbank database? The program used with their options should be reported. The same issue is repeated in line 427.
Line 443. I’m afraid that the word align is not appropriate in this context.
The references are not properly formatted. I’m afraid that all Italics for genes and species are missing.
Summary Review
Although the paper addresses important and interesting questions, the lack of rigor, the unclear framing of questions and the poor organization and presentation of results make it unpublishable currently.
Author Response
The paper by Wu et al. aimed to identify microRNA in wild type and eno2-, a mutant for a key gene involved in glycolysis and other pathways such as those controlling plant growth and abiotic stress responses. In their work, the authors first sequenced small RNA libraries and then they identified potential mRNA targets. Finally, fifteen miRNAs along with their targets were chosen to validate their expression by qRT-PCR.
Although the paper showed interesting results, I found several major concerns that should be addressed:
In my point of view, one of the main concerns is the number of sequenced samples. Indeed, at line 131 the authors reported that only two libraries were produced (one for wild type and one for the mutant). It is not clear to me how the authors can identify differentially expressed miRNAs by using only a single replicate rather than three biological samples that are usually required to avoid spurious results. Which kind of statistics the authors applied to identify differentially expressed miRNA? Parametric tests such as DEseq2?
Response:
Theoretically, there should be at least three biological replicates, but due to the limited funding of the laboratory, there is no duplication in the experiment. Therefore, in our experiment, the software for differential analysis used in this sequencing is edgeR, which can be used to analyze samples without biological duplication. Besides, 15 miRNAs related to seed germination were selected in the test, all of them were verified by qRT-PCR, and the results were completely consistent with sequencing, indicating that the sequencing results were reliable.
In our experiment, we used DEseq2 to identify the differentially expressed miRNAs, and defined the differential miRNA with log2 FC > 2 and P < 0.05 as significance. We have added the information in our text and marked them red.
Another major concern regards the lack of rigor. For example, in Figure 1 is reported the statistical analysis of WT and mutant. However, in the same figure the construct of ENO2- is also shown. I suggest moving it in supplementary material or delete it if not necessary. In addition, in the same figure, I see the standard deviation but it is not clear if the difference between WT and ENO2- in terms of germination rate is statistically different (no p.values are reported). Which kind of statistical test was performed with these data? Please add also the legend for X-axis. Another major concern regards inconsistencies between the results, text and figures. For example, in line 104 the authors reported that 590 miRNAs were differentially expressed but, in the results section, (line 166) they write 591. In addition, in line 155 they reported that:” was 243 and 247 and 232 are in common”. Figure 3 showed different numbers. This is not exhaustive, since other numbers did not match between figures and text.
Response:
Thanks for your kind suggestion, we have moved the construct of ENO2 in supplementary. We have also added the significant difference and the X-axis of the legend. We are sorry for these inconsistencies in our text, we have carefully corrected them and marked red. Thank you for pointing out the problem.
In my point of view table 1 is confounding. What is the difference between exist_miRNA and exist_miRNA_edit? In addition, it is not clear to me what the authors refer to with tags. Do you mean mapped reads that have a match with microRNA? The legend of table 1 is not sufficient. The authors should provide more details to get it self-explaining.
Response:
Exist_miRNA refers to the miRNAs of Arabidopsis that have been included in miRBase. Exist_miRNA_edit refers to the original sequence of miRNAs changes by one base, which leads to the change of target genes.
The small RNA data obtained by preliminary filtering of the original offline data were called clean-reads, and the data obtained by re-filtering this part of data under the following conditions are called clean-tags.
1) Removing low quality reads containing more than one low quality (Q-value≤20) base or
containing unknown nucleotides(N);
2) Removing reads without 3’adapters;
3) Removing reads containing 5’adapters;
4) Removing reads containing 3’ and 5’ adapters but no small RNA fragment between them;
5) Removing reads containing ployA in small RNA fragment;
6) Removing reads shorter than 18nt (not include adapters).
We have also added some details for table 1, which are marked red (line 161-162).
In line 149 the authors reported as follows: “Statistical analysis showed that in WT and eno2-, the number of tag species was 1053660 and 1017006, and the tag abundance was 9661299 and 10071178, respectively (Tab. 1).” Where is the statistical results? Which kind of statistical test the authors applied?
Response:
In the original text, the word “statistical analysis” we used was inappropriate. What we wanted to express here is not a statistical analysis, but the description of the amount of sequencing data, to evaluate the quality of our sequencing. Therefore, this part has been modified in the text, and we also marked them red (line 147-149).
Minor concerns:
I suggest using comma or point when writing numbers in the text
Response:
Thanks for your suggestion, we have checked the whole text.
Line 132. I think these numbers refer to row reads and not to clean reads.
Response:
What we wanted to express here were clean reads. Only after further filtering of clean reads can we got clean tags for subsequent analysis. The specific filtering conditions are as follows:
1) Removing low quality reads containing more than one low quality (Q-value≤20) base or
containing unknown nucleotides(N);
2) Removing reads without 3’adapters;
3) Removing reads containing 5’adapters;
4) Removing reads containing 3’ and 5’ adapters but no small RNA fragment between them;
5) Removing reads containing ployA in small RNA fragment;
6) Removing reads shorter than 18nt (not include adapters).
The authors should be more rigorous in writing gene names. The gene names are not always in Italics (i.e. see line 184, 246-247, 79, 95). This is not exhaustive
Response:
Thank you for reminding us, and we have corrected the gene names which were not italic. All these corrections are marked red.
Line 137-138. “but the expression trend of small RNAs with different lengths in the library is similar”. Where we can see this data? If they are not reported I would suggest adding (i.e. data not reported) or remove it if not indispensable
Response:
This sentence is not rigorous enough, we have removed it in the text.
Figures 10 and 11 should be moved in supplementary.
Response:
Thanks for your suggestion, we have moved figures 10 and 11 in supplementary.
In line 147-148 of my pdf, strange symbols appeared. What is rRNAetc> ?
Response:
In our text, rRNA etc includes rRNA, scRNA, snRNA, snoRNA, and tRNA, which have been described in detail in our manuscript and highlighted in red (line 145-146).
Lines 168-171. I think is more appropriate moving this part in the material and methods section or in the legend
Response:
We have moved this description in the legend of the volcano figure, and we marked it red (line 185-187).
Line 173. What the authors mean when they say: “we mapped”? I think this word is not appropriate
Response:
We have revised the expression approach of the whole sentence, which are marked in red (line 169-171).
Figure 5. Which kind of data the authors used to drawn this plot? Legend is not sufficient, and it is not clear to me if the authors used TPM data or transformed one.
Response:
We use log2 (TPM) to draw the heatmap (Fig. 5), which has been modified in our paper (line 192-193). Thank you for your suggestion.
Line 224:” According to the analysis, 133 target genes of miRNA were enriched in 69 metabolic pathways. Among the 20 enrichment pathways …”. Are the metabolic pathways 69 or 20? Did the authors mean among the 20 top enriched pathways?
Response:
The metabolic pathways are 69, and the second sentence means among the 20 top enriched pathways. We have corrected the expression ways to avoid misunderstandings (line 224).
Figure 8 and 9. Is it possible to apply a statistical test? This could help the readers to understand if the difference in such miRNA expression in WT and eno2- is significant or not.
Response:
Thanks for your kind suggestion, we have added the significant difference using the Student’s t-test (line 256-260; line 265-269).
In the material and methods section many details regarding the company and their city where the reagents were bought are missing. Please add more details in the material and methods section.
Response:
Thank you for the reminder. We have added more details in the material and methods section.
Line 423. How the authors aligned the clean reads to the Genbank database? The program used with their options should be reported. The same issue is repeated in line 427.
Response:
The software Blastall 2.2.25 (BLASTN) was used to compare with the rRNA, scRNA, snoRNA, snRNA, and tRNA in GenBank (Release 209.0) database, and the tags with identity more than 97% were removed . At the same time, the software Blastall 2.2.25 (BLASTN) was also used to compare with the rRNA, scRNA, snoRNA, snRNA, and tRNA in Rfam (Release 11.0) database, and the tags with identity more than 97% were removed. We have also added the details of the method in our text (line 425-431).
Line 443. I’m afraid that the word align is not appropriate in this context.
Response:
We have revised the expression of the whole sentence, and we marked it red (line 455-459).
The references are not properly formatted. I’m afraid that all Italics for genes and species are missing.
Response:
We have checked the references and corrected the genes and species which were not italic. All these corrections are marked red.
Summary Review
Although the paper addresses important and interesting questions, the lack of rigor, the unclear framing of questions and the poor organization and presentation of results make it unpublishable currently.

Reviewer 3 Report
The article titled "Deep sequencing of small RNA reveals the molecular regulatory network of ENO2 regulating seed germination" by Wu et al. investigated regulations of small RNA in wild type and ENO2 mutant of Arabidopsis.
After deep reviewing of the manuscript, I fully recommend it to be published. The results shown are interesting.
Some minor English corrections are needed
Author Response
The article titled "Deep sequencing of small RNA reveals the molecular regulatory network of ENO2 regulating seed germination" by Wu et al. investigated regulations of small RNA in wild type and ENO2 mutant of Arabidopsis.
After deep reviewing of the manuscript, I fully recommend it to be published. The results shown are interesting.
Some minor English corrections are needed
Response:
Thanks for your approval, and we have corrected some English in our text.

Round 2
Reviewer 2 Report
In the revised version of the manuscript, the authors made many improvements. However, I still have some concerns:
- The authors reported that they used DEseq2 (line 452) for differential analysis. I reported below from DEseq2 manual why it is not possible: “If a DESeqDataSet is provided with an experimental design without replicates, a warning is printed, that the samples are treated as replicates for estimation of dispersion. This kind of analysis is only useful for exploring the data, but will not provide the kind of proper statistical inference on differences between groups. Without biological replicates, it is not possible to estimate the biological variability of each gene. More details can be found in the manual page for ?DESeq.”
- The authors reported that they used log2 (TPM) to draw figure 5. Log2 values are used when a comparison is made (control Vs treatment). This procedure help to say that a gene or a miRNA is expressed two fold more in the treated sample compared to the control. How is possible to have two tracks in the figure (one for control and one for ENO2-)?
- As above, the authors reported in figure 4: “The volcano map of DEMs in WT and eno2-.” However, what is expected is DEM in eno2- compared to WT.
- The authors should submit their fastq to such database (e.g, SRA database)
For these reasons, my opinion regarding the manuscript did not change.
Author Response
In the revised version of the manuscript, the authors made many improvements. However, I still have some concerns:
- The authors reported that they used DEseq2 (line 452) for differential analysis. I reported below from DEseq2 manual why it is not possible: “If a DESeqDataSet is provided with an experimental design without replicates, a warning is printed, that the samples are treated as replicates for estimation of dispersion. This kind of analysis is only useful for exploring the data, but will not provide the kind of proper statistical inference on differences between groups. Without biological replicates, it is not possible to estimate the biological variability of each gene. More details can be found in the manual page for ? DESeq.”
Response:
We apologize for this writing error, the DEseq2 could not deal with the data without replicates, so we used edgeR for differential analysis. Both DEseq2 and edgeR can be used for gene differential expression analysis. The same point is that they both process count data and are based on a negative binomial distribution model. Therefore, we will find that most of the genes screened under the same threshold are the same, but some of the differences should be due to the different methods of estimating dispersion. Due to the limited funding in our lab, the experimental design was not repeated, so we selected 18 miRNAs for qRT-PCR validation, and the validation results were consistent with the sequencing results, so the sequencing results of this small RNA were relatively reliable.
- The authors reported that they used log2 (TPM) to draw figure 5. Log2 values are used when a comparison is made (control Vs treatment). This procedure help to say that a gene or a miRNA is expressed two fold more in the treated sample compared to the control. How is possible to have two tracks in the figure (one for control and one for ENO2-)?
Response:
We have corrected this mistake in our final text, thanks for your kindly remind.
- As above, the authors reported in figure 4: “The volcano map of DEMs in WT and eno2-.” However, what is expected is DEM in eno2- compared to WT.
Response:
Based on the volcano map, we wanted to show the miRNAs that were up-regulated, down-regulated, and had no significant changes after the comparison between eno2- and WT as a whole. What was different from Fig. 5 in this map was that the comparison can not clearly show which miRNAs had significant changes. To better display the significantly different miRNAs in Fig. 4, we plotted Fig. 5.
- The authors should submit their fastq to such database (e.g, SRA database)
Response:
Thanks for your kind reminder. We have submitted our fastq data to the SRA database, and the submission number was SUB9500128. We also added this information to our manuscript and marked it red.
